# Semi-Supervised Two-Stage Abdominal Organ and Tumor Segmentation Model with Pseudo-Labeling

Li Mao[1][0000−0002−5383−379X]

AI Lab, Deepwise Healthcare, 100081, Beijing, China
`maoli@deepwise.com`

**Abstract.** In real-world scenarios, such as abdominal organ and tumor segmentation, obtaining complete labels for all classes often presents significant challenges. Additionally, optimizing GPU efficiency emerges as another critical factor in the abdominal organ and tumor segmentation process.

To address the challenge of partial labeling, a semi-supervised approach was employed. Initially, a larger model was trained using complete labels, which was then utilized to generate pseudo-labels. Subsequently, a smaller model was trained on these pseudo-labels. To mitigate GPU memory consumption, a two-stage strategy was implemented. Firstly, an abdomen location model was trained to accurately identify the abdominal area. Subsequently, the segmentation process was restricted to this localized area, thereby reducing the GPU memory requirements.

Experiments on the FLARE23 challenge exhibited promising performance, with an average actual running time of 25.971 seconds, an average AUC-GPU (Area Under the Curve of GPU memory consumption) of 28463.7 MB, and an average maximum GPU memory usage of 2.6 GB on the validation set, and the average running time on the testing set was 18.95 seconds, with AUC-GPU of 20790 MB. Moreover, the model achieved a Dice coefficient (DSC) of 79.99% for organ segmentation and 27.99% for tumor segmentation on the public validation dataset, and 80.67% and 24.02% for the DSC of organ and tumor segmentation on the testing result.

**Keywords:** Segmentation · Partial-label Segmentation · Computational efficiency.

## 1 Introduction

In recent years, there has been an increasing adoption of deep learning-based segmentation models in the field of medicine[1][2][3][4][5]. However, the exorbitant cost entailed in the annotation of medical images has given rise to an increasingly acute dearth of labeled data. Furthermore, owing to the frequent collection of medical data from disparate centers, the predicament of incomplete data labeling has become a pervasive and recurrent concern.

Currently, there have been several studies addressing the training of models on partial-labeled data. Zhang et al. [6] designed a propagated self-training method that further guarantees the quality of the pseudo-labels and improves the richness of the labeled data. Petit et al. [7]introduce an iterative confidence self-training approach inspired by curriculum learning to relabel missing pixel labels. Li et al. [8] proposed the Conditional Dynamic Attention Network (CDANet) that fusing the conditional and multiscale information to better distinguish among different tasks and promoting more attention to task-related features.

On the other hand, reducing GPU utilization is crucial for the practical implementation of the model. More efficient implementations are necessary, as most segmentation methods are computationally expensive, and the amount of medical imaging data is growing[9]. The approach involves the design of efficient models, wherein the architecture is meticulously crafted and developed. ENet [10] serves as an exemplar of this approach. Conversely, anothor approach revolves around network compression, wherein lightweight models such as ICNet [11] are devised, employing pruning methods [12] that are extensively employed in image classification models.

The Fast, Low-resource, and Accurate oRgan and Pan-cancer sEgmentation in Abdomen CT (MICCAI FLARE 2023) is a competition that aims at efficiently segmenting of 13 organs (liver, spleen, pancreas, right kidney, left kidney, stomach, gallbladder, esophagus, aorta, inferior vena cava, right adrenal gland, left adrenal gland, and duodenum) and pan-cancer, i.e., all kinds of cancer types (such as liver cancer, kidney cancer, stomach cancer, pancreas cancer, colon cancer) . The FLARE competition provided a training set includes 4000 3D CT scans from 30+ medical centers. 2200 cases have partial labels and 1800 cases are unlabeled. Despite providing a substantial amount of multi-center data, the dataset suffers from incomplete labeling, necessitating consideration of this issue during model training.

In this paper, we proposed a weakly supervised approach based on pseudo-labeling. Firstly, a large parameterized organ segmentation model was trained, enabling the generation of pseudo labels. Subsequently, leveraging these pseudo labels, a small parameterized segmentation model was trained. To mitigate the GPU memory consumption, an abdomen region location model was trained aimed at minimizing the size of segmentation model input tensor.

## 2    Method

As illustrated in Figure 1, this study begins by training a large parameterized model for organ segmentation (organ segmentation model), then this model is utilized to annotate the remaining data with missing labels. An abdominal region classification model was trained to identify the abdominal slices. Eventually, by leveraging the pseudo-labeled data, we train an efficient lightweight model for both organ and tumor segmentation.

## 2.1   Preprocessing

Before being fed into the 3D U-Net model, a preprocessing step was performed on the CT images to enhance their suitability for analysis. This involved applying z-score normalization, which effectively standardized the pixel intensity values across the dataset. By normalizing the images, any variations in brightness and contrast were minimized, ensuring consistent and reliable input for subsequent processing.

Following the normalization step, an additional resampling procedure was conducted to achieve uniformity in voxel spacing. The images were resampled to a unified spacing of $2 \times 2 \times 3$ for the x, y, and z axes. This adjustment not only facilitated easier comparison and analysis of the data but also eliminated any potential distortions caused by variations in voxel dimensions.

To further optimize the input for the 3D U-Net model, the patch size of the image slices was carefully chosen. Specifically, a patch size of $96 \times 128 \times 160$ was selected for the x, y, and z axes, respectively. This choice ensured that the model received an appropriate and informative region of interest, enabling it to effectively capture relevant features and patterns during the segmentation process.

## 2.2   Proposed Method

.

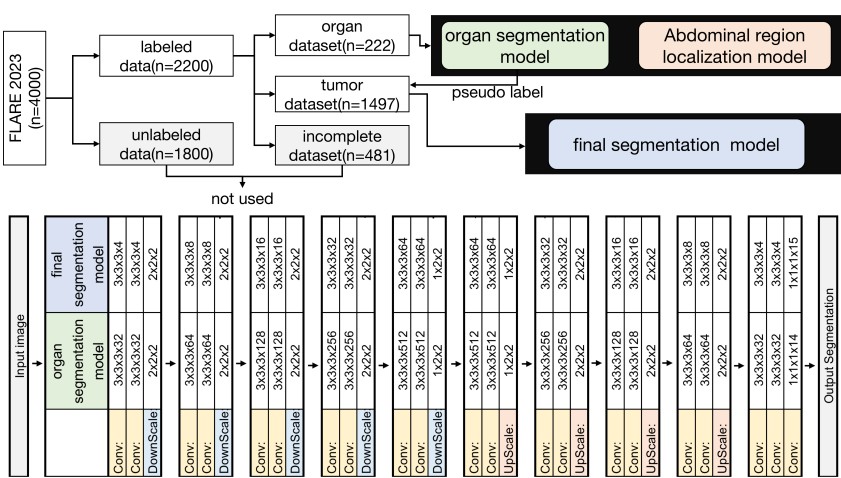

**Fig. 1.** The workflow of our study and the network architecture.

As was shown in Figure 1, we divided the dataset of 2200 cases from MICCAI into three categories: 222 cases with complete organ labels but no tumor labels

(**organ dataset**), 1497 cases with tumor labels but incomplete organ labels (**tumor dataset**), and 481 cases with incomplete organ labels and no tumor labels (**incomplete dataset**). Our pipeline including three models.

The two-stage pipeline in our study was composed of abdominal region localization and abdominal segmentation. Firstly, the abdominal region localization model as employed to identify the abdominal slices. Then, the segmentation model only performed on abdominal region.

### Abdominal region localization model

In abdominal segmentation tasks, certain images incur significant GPU memory consumption. These images encompass non-abdominal regions. This insight motivates us to construct an abdominal region localization model that selectively retains and segments only the abdominal region. The model was trained on organ dataset, as was shown in Figure 1.

In this study, we adopt a mask-based segmentation approach to extract labels that signify the presence of the abdominal region in a given CT image. Specifically, if a CT slice contains a mask, it is classified as representing the abdominal region; conversely, if no mask is present, it is classified as representing a non-abdominal region. Figure 2 illustrate a typical example of the abdominal region and the non-abdominal region.

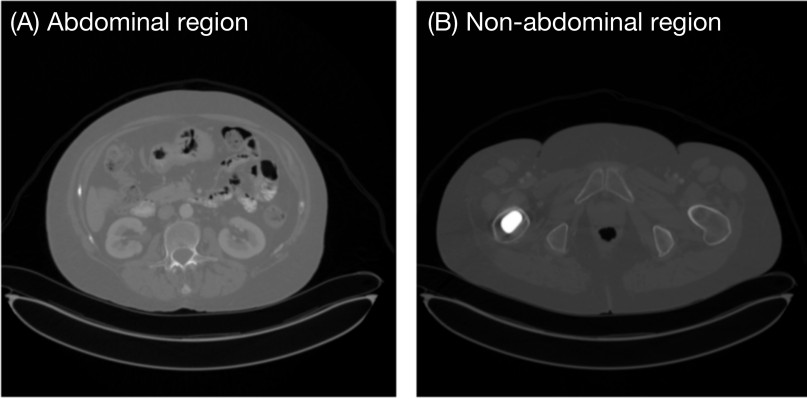

**Fig. 2.** The example of typical (A) abdominal region and (B) Non-abdominal region.

For the organ dataset used in this study, a split of 4/5 and 1/5 was employed for training and testing the Abdominal region localization model, respectively. Initially, the input data undergoes preprocessing steps. The CT images are normalized using a window width of 400 and a window level of 40, enhancing the visibility of soft tissue regions. Subsequently, the images are resampled to a size of $256 \times 256$ and expanded to 3 channels, resulting in a tensor size of $3 \times 256 \times 256$, which serves as the input to the model. The model architecture employed in this study is based on ResNet18 [13].

During the model training process, data augmentation techniques are employed to prevent overfitting and improve generalization. These techniques help in generating additional training samples by applying transformations such as rotations, translations, and scaling to the input data.

Considering that the abdominal localization model itself consumes computational resources, during the inference stage, CT images larger than 50MB in file size are processed layer by layer to obtain predictions for each slice. The maximum and minimum predicted values among all the slices identified as the abdominal region are used to define a continuous region of interest for subsequent segmentation of abdominal organs and cancer region.

### Organ segmentation model

An organ segmentation model was trained on an organ dataset using the nnU-Net framework[14]. The model architecture is based on the 3D U-Net architecture, which consists of 5 downsampling modules and 5 upsampling modules. To improve the accuracy of pseudo label generation, the model has a large number of parameters. Within the nnU-Net framework, the model was trained for a total of 500 epochs with an initial learning rate of 0.01. To achieve robust performance in various medical image segmentation tasks, a compound loss function combining Dice loss and cross-entropy loss was employed, as it has been proven effective [15]. Data augmentation techniques such as mirroring, scaling, rotation, and translation were applied to prevent overfitting.

During training, 4/5 of the organ dataset was used for training the model, while the remaining 1/5 was used for validation. The model with the highest Dice coefficient on the validation set was selected as the optimal organ segmentation model. After training, this model was utilized to annotate a tumor dataset, providing segmentation labels for regions where the tumor dataset was lacking annotations, based on the model's predictions.

### Final segmentation model

To obtain an efficient lightweight model for organ and tumor segmentation, we first reduced the number of parameters of the 3D UNet model. Subsequently, we pretrained the model on the organ dataset using the same training approach as described earlier for the organ segmentation model. After the pretraining procedure, we modified the model's parameters by adding a tensor of the same size to the final output convolutional layer, resulting in 15 output channels representing the background, 13 organs, and tumors. This additional tensor was initialized to zero.

Subsequently, model optimization was performed on the tumor dataset after completing the segmentation label augmentation. A total of 500 epochs were trained with an initial learning rate of 0.01. The loss function, optimizer, and other settings remained consistent with the organ segmentation model. After training, the model with the highest Dice coefficient on the validation set was selected as the final model.

### The remaining dataset

The incomplete dataset and the unlabeled images were not used, as well as any pseudo labels generated by the FLARE21 winning algorithm [16] [17].

### 2.3   Post-processing

To reduce computational load, this study employed GPU-based resampling to ensure that the size of the output segmented images matched that of the original images and corresponded to the abdominal region. No post-processing techniques were employed beyond this to maintain computational efficiency.

## 3   Experiments

### 3.1   Dataset and evaluation measures

The FLARE 2023 challenge is an extension of the FLARE 2021-2022 [18][19], aiming to promote the development of foundation models in abdominal disease analysis. The segmentation targets cover 13 organs and various abdominal lesions. The training dataset is curated from more than 30 medical centers under the license permission, including TCIA [20], LiTS [21], MSD [22], KiTS [23,24], autoPET [25,26], TotalSegmentator [27], and AbdomenCT-1K [28]. The training set includes 4000 abdomen CT scans where 2200 CT scans with partial labels and 1800 CT scans without labels. The validation and testing sets include 100 and 400 CT scans, respectively, which cover various abdominal cancer types, such as liver cancer, kidney cancer, pancreas cancer, colon cancer, gastric cancer, and so on. The organ annotation process used ITK-SNAP [29], nnU-Net [14], and MedSAM [30].

The evaluation metrics encompass two accuracy measures—Dice Similarity Coefficient (DSC) and Normalized Surface Dice (NSD)—alongside two efficiency measures—running time and area under the GPU memory-time curve. These metrics collectively contribute to the ranking computation. Furthermore, the running time and GPU memory consumption are considered within tolerances of 15 seconds and 4 GB, respectively.

### 3.2   Implementation details

**Environment settings** The development environments and requirements are presented in Table 1.

**Training protocols** The training protocals of the final segmentation models can be found in Table 2. As for the abdominal region location model, the optimization process utilizes the Adam optimizer with an initial learning rate of 0.0001. The Cross Entropy Loss function is employed as the objective function for training the model.

## 4   Results and discussion

### 4.1   Performance of abdominal region localization model

The organ dataset composed of 56989 images (17716 labeled non-abdominal image , and 39273 labeled abdominal image), and was splited into training

**Table 1.** Development environments and requirements.

| System | Ubuntu 16.04.3 LTS |
|---|---|
| CPU | Intel(R) Xeon(R) CPU E5-2685 v3 @ 2.60GHz |
| RAM | 16×16GB; (the memory speed is not available ) |
| GPU (number and type) | One TITAN Xp 12G |
| CUDA version | 10.1 |
| Programming language | Python 3.6.8 |
| Deep learning framework | torch 1.6.0, torchvision 0.7.0 |
| Specific dependencies | NA |
| Code | the github link will be provided after acceptance |

**Table 2.** Training protocols.

| Network initialization | InitWeights_He |
|---|---|
| Batch size | 2 |
| Patch size | 96×128×160 |
| Total epochs | 500 |
| Optimizer | Adam |
| Initial learning rate (lr) | 0.01 |
| Lr decay schedule | Polynomial Learning Rateschedule |
| Training time | 11.25 hours |
| Loss function | $0.5 \times L_{BCE} + 0.5 \times L_{Dice}$ |
| Number of model parameters | 1.385M[1] |
| Number of flops | 34.185G[2] |
| $CO_2$eq | Kg[3] |

set(13585 labeled non-abdominal image , and 31078 labeled abdominal image) and testing set(4131 labeled non-abdominal image , and 8195 labeled abdominal image). On the testing set, the abdominal region localization model reached a area under the receiver operating characteristic curve (AUC) of 0.994, accuracy of 0.959, sensitivity of 0.954, and specificity of 0.969.

### 4.2   Segmentation efficiency results on validation set

The efficiency performance on validation set can be found in Table 3. In our approach, we leveraged abdominal localization models to enhance segmentation efficiency in case 0019, 0099, 0063, 0048, and 0029. We observed that the running time was shorter in case 0001 and 0019 compared to the other cases. Additionally, the maximum GPU usage and total GPU usage were also lower in these two cases. In case 0019, for example, out of the total 215 slices, the abdominal region predicted by the abdominal region localization model only spanned from slice 106 to slice 214. This means that only 108 slices were available for segmentation, which resulted in a reduced workload and potentially improved the efficiency of GPU utilization.

**Table 3.** Quantitative evaluation of segmentation efficiency in terms of the running them and GPU memory consumption. Total GPU denotes the area under GPU Memory-Time curve. Evaluation GPU platform: NVIDIA QUADRO RTX5000 (16G).

| Case ID | Image Size | Running Time (s) | Max GPU (MB) | Total GPU (MB) |
|---------|------------|------------------|--------------|----------------|
| 0001 | (512, 512, 55) | 18.85 | 1796 | 16313 |
| 0051 | (512, 512, 100) | 28.21 | 2454 | 34003 |
| 0017 | (512, 512, 150) | 36.33 | 2946 | 47435 |
| 0019 | (512, 512, 215) | 16.54 | 2470 | 17208 |
| 0099 | (512, 512, 334) | 42.7 | 2992 | 27054 |
| 0063 | (512, 512, 448) | 20.15 | 3444 | 24857 |
| 0048 | (512, 512, 499) | 23.07 | 4090 | 31461 |
| 0029 | (512, 512, 554) | 23.83 | 4222 | 33770 |

### 4.3   Quantitative results on validation set

Figure 3 presents four samples of the segmentation results. The first two rows depict well-segmented slices, whereas the last two rows exhibit unsatisfactory segmentation outcomes. In case 0073, the liver was not fully recognized, and in case 0025, the presence of vessels within the liver misled the model.

The quantitative performance can be found in tabel 4. The model proposed in this paper demonstrated favorable performance on organs such as the liver, kidney, spleen, and stomach, achieving Dice Similarity Coefficient (DSC) and Normalized Surface Dice (NSD) scores of over 85% on the validation set. However, the segmentation results for tumors were less satisfactory, with a DSC of

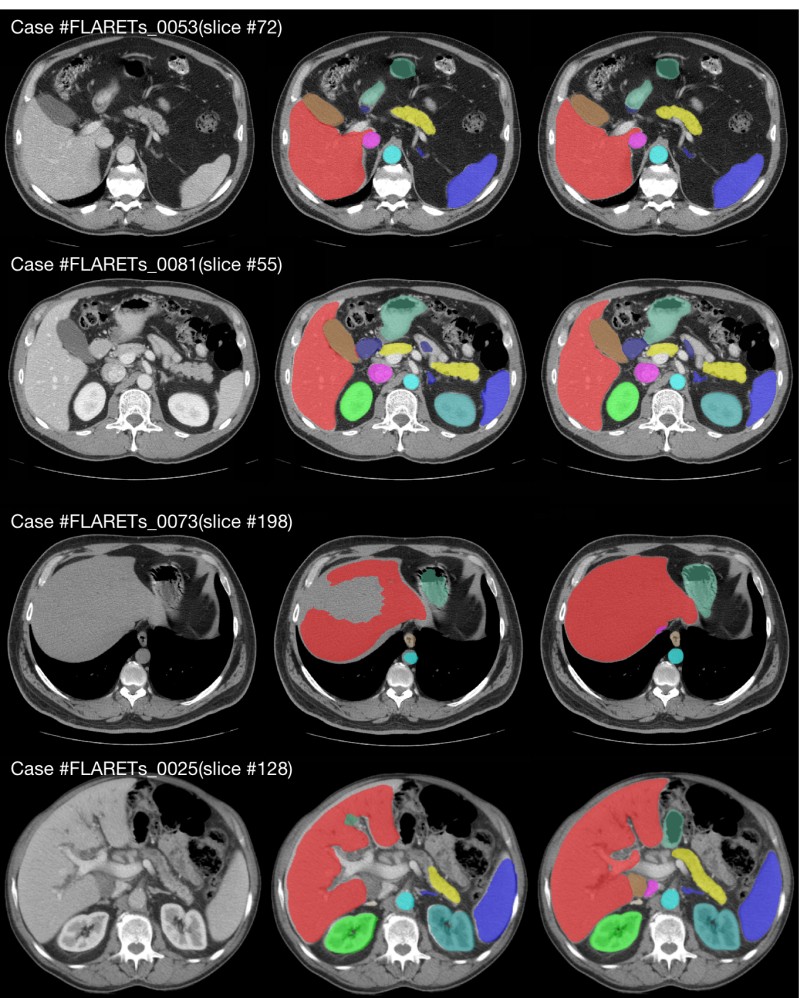

**Fig. 3.** The sample of four cases in validation set.

only 27.99%. This lower performance could be attributed to the significant intra-class heterogeneity and smaller volumes of tumors. The same trend can be found in online validation result.

**Table 4.** Quantitative evaluation results.

| Target | Public Validation | | Online Validation | | Testing | |
|---|---|---|---|---|---|---|
| | DSC(%) | NSD(%) | DSC(%) | NSD(%) | DSC(%) | NSD (%) |
| Liver | 95.43 ± 3.08 | 95.59 ± 5.17 | 97.32 | 98.75 | 93.23 | 93.09 |
| Right Kidney | 87.74 ± 18.95 | 86.46 ± 19.52 | 93.34 | 94.54 | 90.68 | 88.79 |
| Spleen | 93.53 ± 4.16 | 92.92 ± 7.03 | 97.00 | 99.06 | 91.78 | 91.79 |
| Pancreas | 75.05 ± 10.22 | 88.7 ± 9.2 | 83.70 | 95.47 | 73.97 | 86.21 |
| Aorta | 87.36 ± 9.29 | 85.63 ± 15.1 | 94.91 | 98.40 | 90.25 | 91.69 |
| Inferior vena cava | 80.42 ± 15.29 | 77.72 ± 18.02 | 92.03 | 95.26 | 81.87 | 79.13 |
| Right adrenal gland | 68.44 ± 16.59 | 81.89 ± 17.24 | 80.58 | 94.40 | 67.21 | 80.15 |
| Left adrenal gland | 64.27 ± 21.62 | 76.07 ± 24.16 | 80.21 | 93.82 | 65.09 | 76.37 |
| Gallbladder | 74.2 ± 25.74 | 70.63 ± 28.05 | 79.80 | 79.64 | 71.71 | 69.47 |
| Esophagus | 72.06 ± 16.93 | 83.29 ± 16.86 | 82.18 | 93.86 | 79.03 | 91.02 |
| Stomach | 85.65 ± 11.25 | 87.6 ± 13.3 | 92.61 | 96.60 | 86.05 | 88.91 |
| Duodenum | 68.11 ± 15.2 | 86.36 ± 11.48 | 81.92 | 94.46 | 69.74 | 86.33 |
| Left kidney | 87.59 ± 18.6 | 86.33 ± 19.85 | 93.30 | 94.56 | 89.72 | 89.01 |
| Tumor | 27.99 ± 36.29 | 18.17 ± 25.31 | 39.78 | 31.11 | 24.02 | 12.19 |
| Average | 76.27 ± 24.35 | 79.81 ± 25.40 | 88.38 | 94.52 | 80.67 | 85.45 |

### 4.4   Ablation study

The performance of the organ segmentation model is elaborated in Table 5. The DSC and NSD of the organ segmentation model was inferior than the final segmentation model. The average DSC and NSD performance of the organic region was 82.20 % and 86.52 %, lower than that of final segmentation model (88.38% and 94.52

### 4.5   Results on final testing set

The average actual running time on final testing set is 18.95 seconds, with an average AUC-GPU (Area Under the Curve of GPU memory consumption) of 20790 MB. Our model reached a DSC of 80.67% and a NSD of 85.45% for the segmentation of organs. However, the tumor segmentation was relative hard, resulting in a DSC of 24.02% and a NSD of 12.19%.

### 4.6   Limitation and future work

The objective of this study is to minimize GPU memory usage while ensuring accurate prediction outcomes. The average execution time and AUC-GPU

**Table 5.** Organ segmentation model performance.

| Target | Online Validation | |
|---|---|---|
| | DSC(%) | NSD(%) |
| Liver | 93.32 | 92.34 |
| Right Kidney | 86.83 | 86.72 |
| Spleen | 93.42 | 93.12 |
| Pancreas | 77.65 | 89.86 |
| Aorta | 94.83 | 96.63 |
| Inferior vena cava | 91.83 | 92.94 |
| Right adrenal gland | 70.34 | 87.48 |
| Left adrenal gland | 72.79 | 86.80 |
| Gallbladder | 72.09 | 67.73 |
| Esophagus | 76.92 | 89.64 |
| Stomach | 81.17 | 78.37 |
| Duodenum | 70.45 | 79.11 |
| Left kidney | 86.90 | 83.97 |
| Average | 82.20 | 86.52 |

achieved notable efficiency, measuring 18.95 seconds and 20790 MB, respectively, on the testing set. However, compared with the organ region, the segmentation performance of tumor region was relatively low.

The ablation study compared the organ segmentation model and the final segmentation model.In the online validation set, the organ segmentation model exhibited inferior performance compared to the final segmentation model, even in the organic region. The difference indicated that the final segmentation model, trained on the pseudo-labeled dataset, benefited from the organ segmentation model.

This study still have some limitations. Firstly, only two subsets from the FLARE 2023 dataset was used for modeling. During the restriction of deadline, the remaining data was not used. In our future study, the utilization of these data can be performed to enhance the model performance. Secondly, the semi-supervised algorithm in our study is primitively. The state-of-the-art semi-supervised algorithm can be used in our future study. Finally, the pseudo label from the FLARE 2022 models was not used in our study.

## 5    Conclusion

This paper adopts a straightforward and simple approach for partially labeled data segmentation. The organ segmentation model is trained using fully labeled data with complete organ labels. Subsequently, this model is utilized to augment the missing organ labels in the data with tumor labels. This process enables the training of a comprehensive organ-tumor segmentation model. The utilization of the abdominal localization model further enhances the prediction efficiency of the model.

**Acknowledgements** The authors of this paper declare that the segmentation method they implemented for participation in the FLARE 2023 challenge has not used any pre-trained models nor additional datasets other than those provided by the organizers. The proposed solution is fully automatic without any manual intervention. We thank all the data owners for making the CT scans publicly available and CodaLab [31] for hosting the challenge platform.

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

**Table 6.** Checklist Table. Please fill out this checklist table in the answer column.

| Requirements | Answer |
| --- | --- |
| A meaningful title | Yes |
| The number of authors ($\leq 6$) | 1 |
| Author affiliations, Email, and ORCID | Yes |
| Corresponding author is marked | Yes |
| Validation scores are presented in the abstract | Yes |
| Introduction includes at least three parts: background, related work, and motivation | Yes |
| A pipeline/network figure is provided | 1 |
| Pre-processing | 2 |
| Strategies to use the partial label | 4 |
| Strategies to use the unlabeled images. | 5 |
| Strategies to improve model inference | 4 |
| Post-processing | 5 |
| Dataset and evaluation metric section is presented | 5 |
| Environment setting table is provided | 6 |
| Training protocol table is provided | 6 |
| Ablation study | 10 |
| Efficiency evaluation results are provided | Table 3 |
| Visualized segmentation example is provided | 2 |
| Limitation and future work are presented | Yes |
| Reference format is consistent. | Yes |