# OpenReview forum: "Semi-Supervised Two-Stage Abdominal Organ and Tumor Segmentation Model with Pseudo-Labeling"
_MICCAI.org/2023/FLARE — Submitted to FLARE 2023_

### Official Review · Reviewer_TXeV · 2023-09-21
**Semi-Supervised Two-Stage Abdominal Organ and Tumor Segmentation Model with Pseudo-Labeling**

**Rating:** 3
**Confidence:** 5

**Review:**

Summary:

The author proposes a two-stage segmentation approach, which can reduce the inference time of the model. However, regrettably, the completeness of this article is not satisfactory.

Comments:

1、The author did not utilize unlabeled data in this study. If they had done so, the metrics could have been further improved.

2、The author did not provide a detailed description of the specific processes involved in the two-stage segmentation.

---

> ### Author Response · Authors · 2023-11-14
> **Point-by-point response**
>
> 1. The author did not utilize unlabeled data in this study. If they had done so, the metrics could have been further improved.
>
> Thank you for your comment. The partial labeled has been used in our paper (named "tumor dataset"  in manuscript ). The unlabeled data was not used in our study, and was described in limitation, “Firstly, only two subsets from the FLARE 2023 dataset was used for modeling. During the restriction of deadline, the remaining data was not used.”
>
> 2.The author did not provide a detailed description of the specific processes involved in the two-stage segmentation.
>
> Thank you for your comment.  I have revised the method section and add more detailed information about the two-stage segmentation, as well as the performance of the abdominal region localization model. Besides, I have added a new section: 4.1Performance of abdominal region localization model.
>
> The two-stage pipeline was also elaborated: "The two-stage pipeline in our study was composed of abdominal region localization and abdominal segmentation. Firstly, the abdominal region localization model as employed to identify the abdominal slices. Then, the segmentation model only performed on abdominal region."

---

### Official Review · Reviewer_XReZ · 2023-09-21
**Missing content and template errors**

**Rating:** 3
**Confidence:** 5

**Review:**

This paper uses pseudo-labels to complete the data in order to solve the partial label problem. And utilize a two-stage segmentation network to Improve the speed of reasoning.

Comments:
1. There are a large number of references that need to be supplied.
2. The description of the method is too rough.
3. Lack of a lot of experimental content.
4. The template prompt content has not been deleted.

---

> ### Author Response · Authors · 2023-11-14
> **Point-by-point response**
>
> 1.There are a large number of references that need to be supplied.
>
> Thank you for your comment. The introduction section was revised for more readable and the references were cited in latex format.
>
> 2.The description of the method is too rough.
>
> Thank you for your comment. I have revised the method section and add more detailed information about the two-stage segmentation, as well as the performance of the abdominal region localization model.
>
> 3.Lack of a lot of experimental content.
>
> Thank you for your comment.  I have revised the method section and add more detailed information about the two-stage segmentation, as well as the performance of the abdominal region localization model.  Besides, I have added a new section: 4.1Performance of abdominal region localization model. The table of model performance was now finished.
>
> 4. The template prompt content has not been deleted.
>
> Thank you for your comment.  I have checked the full manuscript and revised as advised.

---

### Official Review · Reviewer_89oW · 2023-09-26

**Rating:** 5
**Confidence:** 5

**Review:**

This paper introduces a two-stage segmentation model for abdominal organs and tumors. The main contribution of the proposed method is integrating existing segmentation models to handle weakly-labeled data. Despite its a bit lack of novelty, its substantial engineering practice is impressive.

However, I still have a number of major concerns:

1.  The writing of this paper is completely unqualified for publication. Please authors carefully check the writing overall, for example, a few questions are as follows:
   - References are not marked in related work.
   - Fig. 1 is not clear enough.
   - Parts of discussion, limitations, and future work also should be written seriously. These are not useless parts.
   - Even the submission title on the OpenReview is written as the missing 'S' version of 'Semi-supervised': 'emi-Supervised'.
2.  How's the segmentation performance of the first stage network? And how's the performance of the vanilla segmentation model without the proposed method? More results and discussion should be given to prove the effectiveness of the proposed method.

---

> ### Author Response · Authors · 2023-11-14
> **Point-by-point response**
>
> 1. The writing of this paper is completely unqualified for publication. Please authors carefully check the writing overall, for example, a few questions are as follows:
> (1)References are not marked in related work.
> (2)Fig. 1 is not clear enough.
> (3)Parts of discussion, limitations, and future work also should be written seriously. These are not useless parts.
> (4)Even the submission title on the OpenReview is written as the missing 'S' version of 'Semi-supervised': 'emi-Supervised'.
>
>
> Thank you for your comment.
>
> (1) The introduction section was revised for more readable and the references were cited in latex format.
>
> (2) The figures in our study was reviewed (including Fig.1), and was modified as advised.
>
> (3)Thank you for your professional comment. We have rewrite the limitations for readability, and elaborated the the analysis of the result, including the ablation study.
>
> (4)The title in my PDF is "Semi-Supervised", but the initial letter was missed when submit to the platform. sorry for this mistake.
>
> 2.How's the segmentation performance of the first stage network? And how's the performance of the vanilla segmentation model without the proposed method? More results and discussion should be given to prove the effectiveness of the proposed method.
>
> I have revised the method section and add more detailed information about the two-stage segmentation. We have added the result of abdominal region localization model in a new section: 4.1Performance of abdominal region localization model. The large model performance can be found in the ablation study section: 4.4 Ablation study. The small model was the final segmentation model, the performance has been described in Table 4. Quantitative evaluation results.

---

### Official Review · Reviewer_tUJW · 2023-09-26
**The content is incomplete and does not meet all the requirements of the template.**

**Rating:** 5
**Confidence:** 5

**Review:**

Summary:

This paper utilizes labeled data for training a fine large-scale model, inferences unlabeled data and supplements missing organ labels. Subsequently utilizes pseudo-labeled datasets to train lightweight models and result in good inference performance. However, the content of the article is incomplete and does not meet the template requirements provided by the official.

Comments:

1. Fig. 1 is not clear enough and the font in Fig. 2 is too small.
2. Discussion sections and limits and future work should be more complete.
3. The experiment should have been more complete, showing more details of large and small models, including comparison of large and small models.
4. The specific process involved in two-stage segmentation needs to be described in detail.

---

> ### Author Response · Authors · 2023-11-14
> **Point-by-point response**
>
> 1.Fig. 1 is not clear enough and the font in Fig. 2 is too small.
>
> Thank you for your comment. The figures in our study was reviewed (including Fig.1 and Fig.2), and was modified as advised.
>
> 2. Discussion sections and limits and future work should be more complete.
>
> Thank you for your professional comment.Thank you for your professional comment. We have rewrite the limitations for readability, and elaborated the the analysis of the result, including the ablation study.
>
> 3. The experiment should have been more complete, showing more details of large and small models, including comparison of large and small models.
>
> Thank you for your professional comment. We have added the result of abdominal region localization model in a new section: 4.1Performance of abdominal region localization model. The large model performance can be found in the ablation study section: 4.4 Ablation study. The small model was the final segmentation model, the performance has been described in Table 4. Quantitative evaluation results.
>
> 4. The specific process involved in two-stage segmentation needs to be described in detail.
>
> Thank you for your professional comment.  I have revised the method section and add more detailed information about the two-stage segmentation, as well as the performance of the abdominal region  localization model. The two-stage pipeline was also elaborated: "The two-stage pipeline in our study was composed of abdominal region localization and abdominal segmentation. Firstly, the abdominal region localization model as employed to identify the abdominal slices. Then, the segmentation model only performed on abdominal region."

---

### Official Review · Reviewer_2Ex4 · 2023-09-30
**Missing experiment information in the paper.**

**Rating:** 4
**Confidence:** 4

**Review:**

Pros:
1. The proposed method achieves accurate and efficient segmentation of abdominal organs and tumors. In online validation, the DSC values for organ and tumor segmentation are 79.99% and 27.99%, respectively. The inference time was an average of 25.971 seconds and the area under the GPU memory time curve was an average of 28463.7 MB.

Cons:
1. Title: "emi-Supervised" should be "Semi-Supervised".
2. It would be highly beneficial if the authors could provide open-source code.
3. Abstract: Please add "MB" after "28463.7". Please replace "2.6G" with "2.6GB", and "test set" with "validation set".
4. Fig.1: Please enlarge the text. Please add more information in the figure annotation to explain the process of this pipeline.
5. Abdominal region localization model
6. Section 3.1:  "aiming to aim to" should be "aiming to".
7. Table 3: Please add information about Online Validation.
8. Fig.2: Please enlarge the text.
9. Experiments: More details of the experiment should be presented, e.g. ablation study.

---

> ### Author Response · Authors · 2023-11-07
> **Point-by-point response**
>
> 1. Title: "emi-Supervised" should be "Semi-Supervised".
>
> Thank you for your comment. The title in my PDF is "Semi-Supervised", but the initial letter was missed when submit to the platform. sorry for this mistake.
>
> 2.It would be highly beneficial if the authors could provide open-source code.
>
> Thank you for your comment. The open-source code can be available on github after accept. The segmentation model is based on open-source nnUNet framework. Thus my open-source code will include the abdominal region localization model and the inference pipeline.
>
> 3.Abstract: Please add "MB" after "28463.7". Please replace "2.6G" with "2.6GB", and "test set" with "validation set".
>
> Thank you for your professional comment. I have modified my manuscript as advised.
>
> 4.Fig.1: Please enlarge the text. Please add more information in the figure annotation to explain the process of this pipeline.
>
> Thank you for your professional comment. I have modified my manuscript as advised. The Fig. 1 has been modified and add more annotation.
>
> 5.Abdominal region localization model
>
> Thank you for your comment. We have added this information in a new section: 4.1Performance of abdominal region localization model. Besides, In the method section, the Abdominal region localization model was elaborated.
>
> 6.Section 3.1: "aiming to aim to" should be "aiming to".
>
> Thank you for your professional comment. I have modified my manuscript as advised.
>
> 7.Table 3: Please add information about Online Validation.
>
> Thank you for your comment. Sorry for the missing of this information. I have add it as well as the Testing result.
>
> 8. Fig.2: Please enlarge the text.
>
> Thank you for your professional comment. I have modified my manuscript as advised.
>
> 9. Experiments: More details of the experiment should be presented, e.g. ablation study.
>
> Thank you for your professional comment. A new section (4.4 Ablation study ) has been added. Besides, we have added this information in a new section: 4.1Performance of abdominal region localization model. In the method section, the Abdominal region localization model was elaborated.

---

### Official Review · Reviewer_zDhi · 2023-10-04
**The description of the method in the paper is somewhat vague and could benefit from more detailed explanations**

**Rating:** 5
**Confidence:** 4

**Review:**

Summary:

The method achieves precise abdominal organ and tumor segmentation during online validation, with DSC values of 79.99% for organs and 27.99% for tumors. It's also efficient, with an average inference time of 25.971 seconds and an average GPU memory-time curve area of 28463.7 MB.

Comments for improvements:

Consider changing the title "Semi-Supervised"

Improve abstract clarity: Add "MB" after "28463.7," replace "2.6G" with "2.6GB," and change "test set" to "validation set."

Enhance Fig.1 with larger text and more detailed annotations.

In the discussion sections, provide comprehensive insights on limitations, and future work, and consider conducting an ablation study for depth.

Address writing issues, adhere to template requirements, and proofread the content.

Correctly mark references in the related work section for proper citation.

Include more details on the first-stage network's performance and compare it to the vanilla segmentation model.
Clarify and elaborate on the two-stage segmentation process for better understanding.

---

> ### Author Response · Authors · 2023-11-07
> **Point-by-point response**
>
> 1.Consider changing the title "Semi-Supervised"
>
> Thank you for your comment. The title in my PDF is "Semi-Supervised", but the initial letter was missed when submit to the platform. sorry for this mistake.
>
> 2. Improve abstract clarity: Add "MB" after "28463.7," replace "2.6G" with "2.6GB," and change "test set" to "validation set."
>
> Thank you for your professional comment.  I have modified my manuscript as advised.
>
> 3.Enhance Fig.1 with larger text and more detailed annotations.
>
> Thank you for your professional comment.  I have modified my manuscript as advised.
>
> 3.In the discussion sections, provide comprehensive insights on limitations, and future work, and consider conducting an ablation study for depth.
>
> Thank you for your professional comment.  We have rewrite the limitations for readability, and elaborated the the analysis of the result, including the ablation study.
>
> 4.Address writing issues, adhere to template requirements, and proofread the content.
>
> Thank you for your comment. I have proofread and revised the entire text, organized the logic, and supplemented any parts that do not conform to the template.
>
> 5.Correctly mark references in the related work section for proper citation.
>
> Thank you for your comment. We have reviewed the references in our article. Now the reference was cited properly in latex format.
>
> 6.Include more details on the first-stage network's performance and compare it to the vanilla segmentation model. Clarify and elaborate on the two-stage segmentation process for better understanding.
>
> Thank you for your comment. Sorry for the missing of this details. We have added this information in a new section:
> 4.1Performance of abdominal region localization model. Besides, In the method section, the Abdominal region localization model was elaborated. The performance of the organ segmentation model was also add in section 4.4 Ablation study.
>
> The two-stage pipeline was also elaborated: "The two-stage pipeline in our study was composed of abdominal region localization and abdominal segmentation. Firstly, the abdominal region localization model as employed to identify the abdominal slices. Then, the segmentation model only performed on abdominal region."

---

### Decision · Program_Chairs · 2023-10-24

Accept

---

> ### Author Response · Authors · 2023-11-14
> **Revised manuscript has been uploaded**
>
> 1. Each reviewer's comments has been responded  point-by-point .
>
> 2.The revised manuscript as well as  a Zip file of copyright and latex source file have been uploaded.